# Integrated Electrochemical Biosensors for Detection of Waterborne Pathogens in Low-Resource Settings

**DOI:** 10.3390/bios10040036

**Published:** 2020-04-13

**Authors:** Joshua Rainbow, Eliska Sedlackova, Shu Jiang, Grace Maxted, Despina Moschou, Lukas Richtera, Pedro Estrela

**Affiliations:** 1Centre for Biosensors, Bioelectronics and Biodevices (C3Bio) and Department of Electronic and Electrical Engineering, University of Bath, Bath BA2 7AY, UKgmm46@bath.ac.uk (G.M.); d.moschou@bath.ac.uk (D.M.); 2Department of Chemistry and Biochemistry, Mendel University in Brno, Zemědělská 1665/1, 613 00 Brno, Czech Republic; eliska.sedlackova@mendelu.cz (E.S.); richtera@mendelu.cz (L.R.); 3Central European Institute of Technology, Brno University of Technology, Purkynova 123, 612 00 Brno, Czech Republic

**Keywords:** electrochemical biosensors, low-resource settings, point-of-care, in-situ monitoring, microbial pollution, low and middle-income countries (LMICs)

## Abstract

More than 783 million people worldwide are currently without access to clean and safe water. Approximately 1 in 5 cases of mortality due to waterborne diseases involve children, and over 1.5 million cases of waterborne disease occur every year. In the developing world, this makes waterborne diseases the second highest cause of mortality. Such cases of waterborne disease are thought to be caused by poor sanitation, water infrastructure, public knowledge, and lack of suitable water monitoring systems. Conventional laboratory-based techniques are inadequate for effective on-site water quality monitoring purposes. This is due to their need for excessive equipment, operational complexity, lack of affordability, and long sample collection to data analysis times. In this review, we discuss the conventional techniques used in modern-day water quality testing. We discuss the future challenges of water quality testing in the developing world and how conventional techniques fall short of these challenges. Finally, we discuss the development of electrochemical biosensors and current research on the integration of these devices with microfluidic components to develop truly integrated, portable, simple to use and cost-effective devices for use by local environmental agencies, NGOs, and local communities in low-resource settings.

## 1. Global Water Security

Water covers approximately 71% of the Earth’s surface, making the Earth the blue planet of our solar system with the most accessible water [1]. Only 3% of this water comes from freshwater sources and around 0.0001% of the freshwater can be conveniently abstracted by humans, e.g., from lakes, rivers, etc. [2]. Some obvious modern uses of freshwater in the developing world consist of drinking, cooking, washing, heating/cooling, removal of waste, irrigation, energy production, manufacturing, and transportation [3]. Freshwater is also important for public recreation and arguably is vital for mental wellbeing as ‘blue spaces’ to humans across the developed and developing world [4]. The modern-day water crisis sees 783 million people worldwide without access to clean and safe water, 84% residing within rural areas of developing countries [5]. Approximately 1 in 5 cases of mortality worldwide involving children under the age of 5 are related to some form of water-related disease due to poor water quality or lack of sanitation [6]. It can, therefore, be argued that water should be one of the most highly prioritized natural resources found on (and beneath) the surface of the planet. Waterborne diseases caused by microbial pathogens are commonly spread through contaminated fresh and wastewater sources, e.g., rivers, lakes, and reservoirs. Products from the effects of the diseases caused by these pathogens can be further seen in the wastewaters that flow downstream of human settlements, both rural and urban. Waterborne pathogens such as *Escherichia coli, Shigella dysenteriae, Salmonella typhi, Vibrio cholerae, Campylobacter jejuni, Pseudomonas aeruginosa,* and *Legionella Spp.* are commonly found in both fresh and wastewater sources. 

In terms of wastewater treatment, studies have found that, with the rapidly increased population in urban and rural settlements [7,8], people in low- and middle-income countries (LMICs) are often living with poorly developed infrastructures, such as low sewage coverage [9] and low wastewater treatment capacity [10]. These issues result in excreta and greywater being directly disposed of into drainage or flood canals and, eventually, being discharged into rivers and lakes with no treatment at all or only partial treatment using primary measures. Subsequently, untreated wastewater is reused for irrigation [11], which has a negative impact on not only the consumers of the agricultural products, but also the local farming communities [12]. In some cases, under-maintained freshwater supply pipes with leakages may also be in peril of being contaminated by the neighbouring drainage. Untreated wastewater imposes serious problems for the environment and climate, as well as public health. Due to the absence of adequate treatment, organic chemicals, heavy metals, and microorganisms [13] that began in domestic households or industry are released into the ecosystem as pollutants, further providing waterborne disease-related pathogens with a thriving environment.

According to the World Health Organization, waterborne diseases caused by pathogens lead to approximately 600,000 deaths a year in LMICs, coming in at 1.5 million deaths worldwide. In the developing world, this makes waterborne diseases the second highest cause of mortality, only behind lower respiratory infections [14]. Incidences of waterborne disease-related mortality are highly concentrated in the most remote locations of the world, specifically in areas of Africa, Asia, and South America [15,16]. The reasons for the concentrated effect of disease prevalence are thought to be related to the high levels of poverty found in certain areas of these countries, preventing development of adequate wastewater treatment, freshwater transport infrastructure, and affordable water monitoring systems [17]. Further exacerbating the lack of sanitation is a reduced public understanding of the relationship between hygiene, sanitation, and water pollution, making it hard to present the evidence required by government bodies and policymakers to push regulation and effective legislation for hygiene, research, and infrastructure development. It is also suggested that the rapid urbanization and industrialization observed in developing countries causes significant deterioration of surface water sources. This is due to the larger effluent loads from developing domestic and industrial settlements carrying high nutrient loads and microbial species from surface runoff [18,19]. This runoff further contributes to the development of pathogenic microbial communities with the potential for causing disease [20].

When water sources such as lakes, rivers, and reservoirs become unsafe for consumption due to microbial contamination, the responses of high-income countries and LMICs are widely dissimilar. In high-income countries such as the UK and USA, it is the responsibility of water service providers to ensure that water quality is consistently high. These companies are held to that standard by their customers and the regulatory inspectorates and governments [21]. It is up to water service providers to detect point source water contamination and to implement mitigation strategies. Within LMICs it is often the case that water service providers are not held to the same level of regulation due to less strict levels of legislation regarding health and safety. Due to this lower level of regulation and water service legislation, 785 million people lack basic drinking water services, with 2 million people estimated globally using a source with faecal contamination. In low-income countries 22% of the health care sector lack water service, 21% have no service for adequate sanitation, and 22% have deficit in wastewater management [22].

In the effort of developing devices capable of early warning for public health and improved monitoring of local water-source microbiology, it is imperative that such devices are proficient for the task. Such devices should be capable of high sensitivity and selectivity to certain biomarkers, easy-to-use, resistant to biofouling, and able to give a fast response. For application within LMICs especially, it is vital that the device gives an end-user type response and is cost-effective and capable of measurement on-site or in situ. For this purpose, we believe that electrochemical biosensors could be a promising and robust option for the future of these devices.

## 2. Conventional Techniques for Water Quality Monitoring

For the direct determination of waterborne pathogens, several conventional techniques exist, such as MALDI-MS, PCR, microbiological methods, microscopy, pyrosequencing, DNA microarrays, FISH, and immunology-based methods. All of the above-mentioned techniques come with their own benefits and drawbacks. Unfortunately, there is no standardized method that involves collection and analysis of water samples for every waterborne pathogen of interest. It is particularly difficult to measure every single pathogen simultaneously, creating a challenge for conventional detection methods. Further challenges present themselves for water-based detection, such as physiological differences between groups of pathogens, low pathogen concentration in large volumes of sample source, and difficulty of sample collection. Additionally, molecular assays are under continuous development for individual pathogens in each of the major pathogenic groups (viruses, bacteria, protozoa) [23,24]. 

### 2.1. Mass Spectroscopy

Matrix-assisted laser desorption/ionization mass spectrometry (MALDI–MS) used for direct and rapid analysis of water can differentiate bacterial species and subspecies in clinical microbiology laboratories. This method has some advantages such as inexpensive operation as well as rapid analysis. Moreover, MALDI-MS provides high resolution and reproducibility of analysis and the ability to analyse a large number of isolates at the same time. Another advantage of this method is the very small amount of sample required for analysis. Moreover, the MALDI–MS approach generates unique m/z signatures for different microorganisms due to inherent differences in the cellular proteins expressed by the microbiological species. Additionally, MALDI-MS can be used for a wide range of microorganism detection. On the other hand, a limiting factor of this method is the identification of species of microorganisms that are not included in the database already. However, the mass spectra of the new microbial strains are constantly being added to the database. Other disadvantages include expensive instrumentation and the complicated identification of some phylogenetically related bacterial species or mixed cultures. Additionally, this technique requires expertise for handling and proper analysis.

In a study by Hurst et al. it was shown that MALDI-TOF (time-of-flight) MS offers considerable potential as a detection method for PCR-amplified bacterial DNA. The study showed the detection of 108~ and 168-base sequences obtained from PCR specific to *Legionella* [25]. Moreover, He et al. successfully detected more than 20 species of *Legionella* from clinical isolates [26]. Among others, Donohue et al. used MALDI-MS for the analysis of seventeen species of *Aeromonas*, while mass ions had average values of 6301; 12,160; 12,254, and 13,450 m/z [27]. Furthermore, Erler et al. proposed profiling of the potentially pathogenic strains *Vibrio spp.,* which cause several illnesses in humans [28]. They created the VibrioBase (see Figure 1), which provides free access for detailed risk assessment studies pertaining to pathogenic *Vibrio spp*.

### 2.2. Polymerase Chain Reaction (PCR)

The Polymerase Chain Reaction (PCR) is a simple method based on the amplification of a specific DNA sequence. PCR provides relatively rapid and selective amplification of nucleotide sequences within in vitro conditions. In diagnosis, this assay involves several crucial steps, such as DNA extraction from samples, PCR amplification, and detection of amplicons. When the specific clinical samples contain only a few colony forming units (CFU), each experiment must be carefully designed and performed. The main advantages of PCR are high sensitivity, rapid time, and more reliable analysis than classical cultivation methods. Additionally, PCR is widely, and can be adapted according to the analysis purpose for various nucleic acid types. Nevertheless, PCR has limitations including its very high sensitivity, which in contaminated samples may lead to false-positive results. Further, in the presence of a contaminant that inhibits PCR, this may result in false-negative reactions. Also, the quantification of the sample is commonly difficult and requires a skilled technician for analysis [29].

Multiplex PCR (mPCR) allows detection of multiple relevant genes simultaneously. The benefits include a shorter analysis time and cost-effective analysis due to the nature of upscaling. Unfortunately, the main drawback is the optimization, which is more difficult than with one set of primers like in the original PCR. In general, a maximum of four different target sequences are amplified to ensure adequate sensitivity, specificity, and easy data interpretation. mPCR is often used to identify several bacterial species simultaneously or to detect genes encoding staphylococcal enterotoxins or anti-microbial resistance (AMR) genes. According to the study of Kong et al., the common species in real water samples were *Aeromonas hydrophila*, *Shigella flexneri*, *Yersinia enterocolitca*, *Salmonella typhimurium*, *Vibrio cholerae,* and *Vibrio parahaemolyticus* [30]. The detection limit of this method for the bacterial targets was estimated at 10^0^–10^2^ CFU/mL. On the other hand, mPCR can even be used for the detection of protozoan. For instance, Marangi et al. used mPCR for the detection of *Giardia duodenalis* (*G. duodenalis*) and *Cryptosporidium parvum (C. parvum)* [31]. Of 119 water samples, 28.6% were test-positive for *G. duodenalis* and/or *C. parvum*; of 113 mussel samples, 66.6% were test-positive for *G. duodenalis* and/or *C. parvum* pathogens. The oocysts numbers (per 5 μl of DNA sample) ranged from 10 to 64. In that study, the multiplex assay achieved an efficiency of 100% and an R^2^ value higher than 0.99.

For RNA sequences, amplification is commonly done using reverse transcription PCR (RT-PCR). RT-PCR was developed to provide a quantitative determination of waterborne pathogens. Parshionikar et al. used RT-PCR for the analysis of human enteric viruses in both drinking and environmental water samples. This research group focused on enterovirus (POLIV1: 760 copies), hepatovirus (HAV1: 100 copies), and rotavirus (NORV1: 4100 copies) in real samples [32].

A widely used type of PCR is real-time PCR (qPCR), which offers direct detection and quantification of the PCR products “in real-time” instead of after the end of the reaction [33]. For the determination of PCR products, monitoring of the fluorescence signal is utilized, whereby intensity is permanently sensed and analysed, so amplicons do not need to be detected via electrophoresis. Thus, this technique is simplified and accelerated and may provide a useful tool for point-of-care devices (PoC). Determination of the number of nucleic acid molecules in a sample is used to study the detection of pathogenic microorganisms and viruses, as well as gene expression for AMR. The advantage of the qPCR method is the rapidity without the need of the PCR product detection (results within a few hours), ease of execution, accuracy, and sensitivity. In a study by Bhagwat, water samples were analysed that contained *Salmonella typhimurium (S. typhimurium)*, *Listeria monocytogenes (L. monocytogenes),* and *Escherichia coli* (*E. coli)* O157:H7 [34]. Within the samples, counts of *L. monocytogenes* lower than 1000 cells/mL were observed, in the case of *S. typhimurium,* the predicted level was 1 to 10 cells/mL and for *E. coli* the results were almost the same. 

### 2.3. Microbiological Laboratory Methods

Microbiological cultivation methods, including culturing of target microorganisms within/upon nutrient broth or agar-type mediums and their phenotypic classification, have been used in applied microbiology since the 19th century. These methods aim to easily determine the level of contamination of water, especially drinking water. Although these methods are not used in standard water quality testing, they certainly have a place in water microbiology. The reason for their usage is especially justified in specific cases, including extraordinary situations, such as floods or accidents, or in-field analysis measurements. Cultivation methods have some advantages, such as simplicity, sensitivity, and specific analysis, without false-positive and especially false-negative findings. For orientation determination, methods often used include membrane filtration, liquid medium, fermentation, chromogenic liquid medium, fluorogenic substrate, or combinations thereof [35]. The major disadvantage of using this technique is the time needed for significant results, often requiring samples to be left overnight for significant colony growth for analysis. Additionally, while false-positive readings are rare, they do require the need for a skilled technician.

### 2.4. Microscopy Techniques

Microscopic evaluation is one of the basic procedures in the analysis of the microbial population and plays an important role in water analysis. Microscopy is a flexible technique that allows for rapid detection and can help determine the number of microorganisms in water. In general, analysis of water samples can also be performed by microscopy in different forms such as light, fluorescence, electron microscopy, and flow cytometry. Light microscopy is the oldest type of microscopy. It can be used directly in a laboratory for microorganism identification or as a quantitative method for microbial population estimation [36]. In fluorescence microscopy, the excited fluorochrome is exposed by light at a short wavelength, which is subsequently emitted at a longer wavelength from the fluorochrome. The emission spectrum is transferred to higher wavelengths related to the excitation wave because during fluorescence some of the energy is lost. Excited and emitted wavelengths are checked and divided by suitable filters in the microscope. Another microscopy-based technique, electron microscopy, includes two variations: transmission and scanning microscopy. The transmission electron microscope has enormous magnification and resolution, with detection limits in the nanometre range, and allows detailed imaging of individual intracellular structures of microorganisms. In scanning electron microscopy, the sample is coated with a thin film of metal and is scanned by an electron beam. The technique allows for three-dimensional imaging of the sample surface and can be useful for studying microorganisms in their natural environment [37]. 

A flow cytometer is an instrument that analyses particles (cells, bacteria, nuclei) moving in a liquid stream that intersects a focused, stable, and high-intensity beam of light, usually formed by a laser. The instrument records the characteristics of each cell carried by the current and can analyse thousands of cells in a sample in a second. Particle properties including size, shape, nucleic acid content, presence of surface antigens, ability to reflect or scatter light, and fluorescence can be observed. As an automated method, it offers the following advantages in particular: analysis speeds within minutes, easy preparation, and high throughput of samples. Within a single sample, the flow cytometer analyses many thousands of individual particles, which is not manually possible. An obvious advantage is also the ability to collect data about individual cells. Problems may arise in the detection of pathogens against the background of many cells or particles. If specific fluorescent dyes are available for the selective labelling of selected species of microorganisms, the method is potentially very specific. The obvious disadvantage is the financial demands, especially the investment costs for the acquisition of the flow cytometer. The samples cannot be stored for a long time, and solid samples must first be treated to separate cells from each other. Trained personnel are required to operate the cytometer. Another drawback is the lack of differentiation between viable and dead cells and interference with matrix determinations.

### 2.5. Pyrosequencing

Pyrosequencing was one of the first sequencing methods of the new generation, which began to develop as an alternative to the classical Sanger method. Recently, pyrosequencing and other methods have been coming to the forefront, promising to speed up analysis and reduce cost. Pyrosequencing refers to a series of enzymatic reactions during which the incorporation of individual deoxynucleotide triphosphates (dNTPs) into the synthesized chain, due to released visible radiation, is recorded. The DNA template is incubated with several enzymes, including DNA polymerase and ATP sulfurylase. Incorporation of any of the four dNTPs during the synthesis of the complementary strand of sequenced DNA results in the release of pyrophosphate. This is further converted by ATP sulfurylase to ATP, which is then converted by the enzyme to a light signal captured by a charge-coupled device (CCD) sensor. The remaining unincorporated dNTPs are subsequently removed from the reaction by pyrosis and the whole cycle is repeated. The schematic of the basic reaction can be seen in Figure 2. Pyrosequencing is useful not only in basic research of biological processes but also in applied fields such as disease diagnostics and forensic medicine [38]. Similar to the techniques previously discussed, pyrosequencing comes with notable advantages. First, pyrosequencing has particularly high sensitivity, requiring only 5% tumour loads within cancer diagnostics as compared with Sanger sequencing that usually requires a 20% minimum load. Pyrosequencing is also more cost-effective and time-efficient due to use of less reagents and minimal protocol steps. Notably, pyrosequencing does not require electrophoresis for data readout. However, with this rapid and cost-effective readout comes the disadvantage that it can only analyse short nucleotide sequence lengths of < 200 bp. Secondly, pyrosequencing data analysis is a manual process and thus requires a skilled technician [39].

### 2.6. DNA Microarrays

DNA microarrays are useful quantitative and qualitative techniques that are commonly used for monitoring the whole genome, especially for gene expression under varied growing conditions. This technique can detect specific mutations in DNA sequences and can also be used for monitoring in situ. Hybridization may also be reversed, with unlabelled probes attached to a solid support to which a fluorescently labelled test nucleic acid is added, called reverse hybridization. An example of this type of hybridization is a microchip device. Some of the microarrays can contain 10 to 10,000 ssDNA fragments per glass chip. Thus, several thousands of hybridizations can be accomplished simultaneously [40]. Fluorescence detection after hybridization is ensured by a confocal laser scanner coupled to the analysing computer. The intensity of the fluorescent signal is dependent on the amount of nucleic acid; therefore, it is possible to quantitatively and qualitatively determine specific regions of target molecules in one reaction. Upon overlaying with the test sample, the target DNA will bind to the DNA probe in the event of a positive reaction. The target DNA is labelled with a fluorophore capable of emitting light, thus, binding of the target DNA to the probe results in fluorescence. 

While the high throughput of this technique is an obvious advantage, microchip production is relatively expensive. The microarrays can be used to study gene expression (measured by differences in the number of RNA transcripts of individual samples), compare genes of closely related microorganisms, detect antimicrobial resistance genes, and identify pathogenic bacteria in particular. Furthermore, DNA/RNA chips can be used even for virus detection [41]. Therefore, there have been oligonucleotide-based microarrays developed using the sequences of 16S–23S rDNA internal transcribed spacer regions (ITS) and the gyrase subunit B gene (*gyrB*) as these have been found to be the most prevalent and virulent waterborne pathogens. Twenty-six specific probes were used to detect *Aeromonas hydrophila*, *Vibrio cholerae*, *Legionella pneumophila, Pseudomonas aeruginosa*, *Salmonella spp.*, *Vibrio parahaemolyticus*, *Yersinia enterocolitica*, *Shigella spp.*, *Leptospira interrogans,* and *Staphylococcus aureus*. The results were reliable, reproducible, specific, and highly sensitive, with detection of 10^4^ CFU/mL achieved of each target organism with 100% accuracy [42].

### 2.7. Fluorescence In Situ Hybridization (FISH)

Fluorescence in situ hybridization (FISH) is a method that combines a microscopic method and a molecular biological approach. It is a direct method of detecting bacteria by hybridization with fluorescently labelled gene probes. The 16S rRNA conserved regions are mainly used. Visualization is performed using a fluorescent microscope. The 16S rRNA sequence databases are very extensive; at present, in addition to broad taxonomic groups, narrower groups, sometimes even individual genera and species that are difficult to cultivate, can be detected. The main limitation in practical use is the high detection limit, since the lower working limit is 500 bacterial cells per mL (on the membrane filter) to give a sufficient signal. FISH is a standard analysis that is used to refine additional chromosomal analysis. FISH is often used to locate the specific characteristics of the DNA for the purposes of genetic counselling, medical identification, and species identification. This method can also be used for the detection and localization of specific RNA in cells, tumour cells, and tissue samples. The samples are observed under a fluorescence microscope. This technique involves binding a specific DNA sequence (DNA probe), which is labelled with a fluorescent dye complementary to the examined DNA sequence. At elevated temperatures of 72 to 75 °C, the fibres of the DNA probe and the examined sample are distributed. Then, usually during the subsequent cooling to 37 °C, the probe specifically binds to the examined site, leading to hybridization. Fluorescent labelling of the probe then develops a colour signal in the fluorescence. 

Recently, a dual-colour fluorescence in situ hybridization (FISH) assay was developed for the analysis of *Mycobacterium*. The limit of detection for *Mycobacterium tuberculosis* was determined to be 5.1 × 10^4^ CFU/mL and for *Mycobacterium avium,* 1.5 × 10^4^ CFU/mL [43]. Moreover, Almeida at al. [44] proposed a novel method for detection of *Salmonella ssp.* by using a novel approach based on peptide nucleic acid (PNA). The PNA-FISH method was able to determine 1 CFU/10 mL of sample (5 × 10^9^ ± 5x10^8^ CFU/mL after an overnight enrichment step) and 1 CFU/10 g of PIF (powdered infant formula) (2 × 10^7^ ± 5 × 10^6^ CFU/mL after an 8 h enrichment step). Additionally, the probe distinguished the mixed bacterial contamination in wastewater by using 4’,6-diamidino-2-phenylindole (DAPI) as a dye.

### 2.8. Immunological-Based Methods

Immunological-based methods are generally based on a highly specific reaction between the antibody and antigen. This technique has been commonly applied for the determination of specific whole-cell microorganisms, proteins, and toxins. The most common methods for detection of specific microorganisms are immunofluorescence, the enzyme-linked immunosorbent assay (ELISA), and serum neutralization tests (SNT). SNT tests have been used for the serotyping of viruses and involve mixing a sample, extracted from a plaque assay, with antiserum and then assessing the decrease in infectivity by plaque assay. Jain et al. developed a selective, highly specific, and sensitive solid-phase sandwich ELISA procedure for detection of the highly vicious waterborne pathogen *S. typhi* with modified isopore polycarbonate (PC) black membranes. These membrane methods based on immunomagnetic separation (IMS) and immunofluorescence are mostly used to determine protozoan in water and soil samples. The surface of the PC membrane was modified with polyclonal somatic ‘O’ type antibodies (Abs) against whole-cell *S. typhi*. Without any pre-treatment steps, a detection limit of 2 × 10^3^ CFU/mL was achieved, compared with the detection limit of 10^6^–10^7^ CFU/mL by conventional ELISA method [45]. 

## 3. Challenges of Monitoring Fresh and Wastewater Quality in Low-Resource Settings

Global freshwater shortage and pollution of water resources by pathogenic sources are steadily increasing issues. Thus, it is of great importance to improve our methods for water quality monitoring to mitigate the increase of the global disease burden. Current approaches to water quality monitoring suffer from several disadvantages that make them ill-suited for monitoring within low-resource settings. Current methodologies include techniques such as microbiological culturing and colony counting, mass spectroscopy (MS), polymerase chain reaction (PCR), pyrosequencing, microarrays, etc. These monitoring approaches are limited by their difficulty of use, high costs (both monetary and in time), lack of quantitative data, among other issues. These limitations are discussed below as well as the pressures for water quality monitoring requirements.

### 3.1. Ease of Use

Modern-day technology can be non-intuitive to use and therefore requires specialized training and expertise on daily use and device troubleshooting and maintenance. This includes the use of both the physical device itself and the software commonly used for its operation. This kind of training and expertise is usually found in higher education backgrounds and is both time-consuming to carry out and costly to obtain. Therefore, it is rare that a qualified individual is available in low-resource settings to operate such technology. Many of the techniques used in laboratories also follow complicated and exact protocols that also use specific reagents. Understanding of these reagents is crucial to experimental success for detection of target analytes. It is also understood that of the 7.7 billion inhabitants across the world, approximately 4.2 billion are still offline, with the majority of developing countries having over 50% of the population without access [46]. Technology and software rely upon access to the internet for software updates, troubleshooting, and the initial software download. 

Another limitation to the simplicity of current approaches for water quality monitoring is that the techniques used produce data that require complex analysis and interpretation. Many of the techniques used in identification of waterborne pathogenic contaminants give visual data. For example, microbiological colony culturing and FISH give visual data that are commonly interpreted or verified by trained practitioners to obtain results. A device capable of monitoring microbial contaminants in low-resource settings would need to be fully automated. This is due to the end-users of the device likely being local communities, not-for-profit organizations, and environmental agencies. The potential device technological design should simplify the protocol of sample analysis by integrating sample collection, preparation, and measurement, followed by data fitting and analysis to provide simple to understand results on a consumer-focused display interface.

### 3.2. Cost of Device Manufacture and Implementation

In the age of diagnostics that we now find ourselves in, there is a need for truly cost-effective devices that are fully integrated with current manufacturing standards and protocols. For the detection and monitoring of water pollution, this need has never been so prominent. Affordability is one of the key components of the ASSURED criteria for on-site monitoring devices: affordability, sensitivity, specificity, user-friendliness, rapid analysis capability, operation free of excessive equipment, and portability (deliverable to those who need them) [47,48]. Initial purchases of conventional technologies are often expensive but also come with running costs such as insurance, product maintenance, and ongoing reagent costs. Current approaches and technologies for monitoring and mitigation of pollution by microbial populations result in high costs [49]. 

It was estimated in 2015 that associated economic costs of waterborne-pathogen effects on the water industry were approximately USD 11.35 billion per annum [50]. This is perhaps due to the costs of devices needed for laboratory testing that are enormously expensive, costing up to tens of thousands of dollars. The reason for these high price tags lies in the supply–demand of the instruments. Manufacturing demand is comparably low and therefore cost is high due to the high costs of manufacturing processes and materials. Laboratory instruments are not generally mass-produced in short time periods. Additionally, instruments are made without the built-in obsolescence of the white goods appliance market, meaning they are of high quality and experience a long life [51]. This further influences the producers of scientific instrumentation to increase costs due to the less regular purchase cycle. 

One promising avenue of research for the development of cost-effective sensing and monitoring devices is lab-on-chip (LoC) biosensing technology. This involves the integration of sensor electrodes used in electrochemical biosensors with sustainable and cheap polymers and microfluidics for cheap sensing options. LoC biosensor devices can be produced using cost-effective polymers such as plastics and thin-layer metal electrodes [52]. This already reduces the cost of device production, further reduced with miniaturization, which is a common advantage of electrochemical biosensors [53,54]. Due to miniaturization and low-energy requirements, a significant amount of research has been dedicated to the development of cheap and disposable energy sources for these devices. One such group of energy sources that has seen success is biofuel cells, such as those powered by enzymes or microbes as mediators for the conversion of glucose and oxygen as well as other digestate [55,56]. A manufacturing avenue already exists to produce plastic-based sensing platforms in the form of printed circuit boards (PCB). By using this heavily established industrial process, it is possible to achieve single board costs of USD 0.20–0.52 per cm^2^ depending on layer complexity compared with USD 10-20 per cm^2^ for glass-based chips [57]. These costs can then be greatly reduced through mass production for platforms that promise the ability to create truly cost-effective devices.

### 3.3. In Situ Measurement Capability

A significant challenge denoting the modern era of water quality monitoring is that of in situ monitoring. That is, technology that is suited to the detection of biomarkers at the source, e.g., riverside, lakeside, etc., with minimal sample preparation. This is a particular issue for conventional water quality monitoring methods due to the rural nature of many sources [58]. Specific issues will be discussed including portability, specificity, device power source, sample preparation, etc. As previously discussed from the ASSURED criteria, portability is a key criterion for point-of-care (PoC) diagnostic technology [47,48]. However, conventional techniques do not meet this criterion due to large sizes and weights, high energy requirements from an alternating current (AC) source, and lack of robustness. What conventional techniques for water quality monitoring fall short on, PoC devices and lab-on-chip (LoC) systems seem more than capable of taking up. Electrochemical point-of-care LoC systems promise true portability through high sensitivity, selectivity due to immobilized biomolecular probes, and ease of miniaturization [47,59,60]. This ease of portability is due to the integration of microfluidics, electrode-based sensing platforms, and other microfabricated elements that are common in LoC and PoC platforms. One such device by Sher et al. shows the integration of sensing components with microfluidic pressurized chambers and automated microvalves for micromixing, sample sorting, and preparation (see Figure 3). This allowed for the fabrication of an electronic LoC system with dimensions of less than 10 cm^2^ [61].

Water is an arguably complex sample type, more complex than blood, specifically due to the general consensus behind blood content. It is known that blood has commonly known molecules that can be expected, while water samples can contain a plethora of contaminants that vary from source to source. This can be observed even between wastewater and freshwater samples, with wastewater samples containing a mix of urban effluent such as human or animal excretions, pharmaceuticals, heavy metals, household products, etc. [62,63]. While freshwater samples may contain aggregates, i.e., sediments, microplastics, etc., agricultural excretions, and overland flow of fertilizers and pesticides during heavy rainfall events [64,65].

In the case of laboratory testing for waterborne pathogen analysis, complex techniques need to be used to identify and quantify pathogen samples, which can only be done within the laboratory. This requires a lengthy process of water sample collection, sample pre-treatment, storage, and transfer to the laboratory. This process is not only time consuming but also costly due to processing reagents, labour, transport, and storage [66]. Once samples reach the test laboratory, they either need to be analysed instantly or frozen and stored depending on the nature of the pathogen of interest. One such study by Farkas et al. describes a two-step sample concentration before analysis involving ultrafiltration using tangential flow followed by precipitation using polyethylene glycol (PEG) 6000 to reduce sample size from 1–10 litres to 1–4 mL (see Figure 4) [67]. Within LoC devices, it is possible to integrate components for on-chip sample preparation, e.g., cell lysis, macro- and microfiltration, preconcentration, and micromixers for rapid and cost-effective on-site analysis [58,68]. Thus, negating the requirement for sample preparation and storage for later analysis as these devices can analyse the samples instantaneously.

### 3.4. Real-Time Quantitative Data

While it is currently possible to collect in situ real-time quantitative data in fresh and wastewater systems for a plethora of physicochemical parameters, this is currently a challenge for waterborne pathogens [69,70,71]. At present, there is a lack of literature to back-up the practical concept of real-time data collection for waterborne pathogen populations except in water infrastructure facilities. This is especially predominant in low- and middle-income countries where water treatment facilities if they exist, do not have the extra resources for high-resolution water quality testing [72]. For the collection of real-time qualitative data, devices are required to be placed in situ. This raises several challenges for the design of any in situ monitoring device in aquatic environments. It has been more than a decade since the initial coining of the ASSURED criteria for PoC diagnostics [73]. Since then, a plethora of research has been conducted by research groups across the world for the development of PoC devices with these criteria in mind [74]. However, challenges remain, and some new hurdles have since become relevant. To keep the vigour of the research going and to keep up with technological advances, the group headed by Professor Rosanna Peeling have updated the criteria to REASSURED. These new criteria include real-time connectivity and ease of specimen/sample collection as well as whether the technology is environmentally friendly.

According to Land et al., real-time connectivity consists of the device being wirelessly connected to an external reader or mobile device that powers or analyses reactions and displays results to the end-user [74]. Ease of sample collection means that samples must be taken in a non-invasive way from the patients or the surrounding environment. The addition of environmental friendliness arose from the realization that a large amount of device development was using unsustainable and non-recycled materials. Thus, PoC devices should be fabricated using sustainable and recycled options and should be non-invasive/polluting to the patient and/or environment. Although no conventional techniques exist for in situ real-time microbial water quality monitoring yet, challenges have been identified for their development into biosensors. These challenges include the issue of biofouling, powering the device, real-time connectivity, and environmental friendliness. 

Biofouling is the deposition and accumulating growth of organisms such as algae, fungi, bacteria, and other microorganisms on surfaces to form an unwanted biofilm community. Regarding the development of in situ monitoring devices, biofilms can cause damage to a device by reducing sensitivity through prevention of target molecules binding to sensor/capture surfaces due to surface inactivation [75,76]. Several research pathways are being explored for the development of antifouling options for in situ monitoring equipment. One option is to enclose device sensors in a chamber controlled by a microprocessor such as that done by Kelly et al. [75]. In the device design, a μ-processor controls the opening of the chamber to expose the sensors to the sample environment. After analysis is complete, the chamber is closed off from the environment and biocide salts are dissolved into the solution to create an antifouling environment. 

Another method of mitigating biofouling that has been explored is the surface chemistry involving thiolated zwitterion polymers. Zwitterion polymers can be utilized for their inherent characteristic of inducing hydration shell-like layers due to high affinity for water molecules [77]. These hydration shells could be utilized to prevent the binding of proteins from bio-fouling organisms to device surfaces [78]. A study by Jolly et al. implementing thiolated sulfo-betaine as an antifouling agent found a significant decrease in non-specific signal change compared with other blocking agents (see Figure 5). Gold surfaces blocked with non-zwitterionic 6-mercapto-1-hexanol (MCH) saw a signal increase of 12.5% upon incubation with 100 μM human serum albumin (HSA), compared to <1% change with sulfo-betaine [79], suggesting the promising potential of surface-functionalized zwitterionic polymers as antifouling molecules for in situ devices.

Other promising examples of antifouling research for monitoring within complex bio-fouling environments include three-dimensional antifouling coatings such as cross-linked bovine serum albumin, gold nanowires, and glutaraldehyde [76] and the integration of nanogenerators capable of surface electric double-layer disturbance (see Figure 6) [80]. 

A previously mentioned challenge of implementing conventional techniques in on-site monitoring of aquatic systems is available power sources. Conventional techniques require larger energy inputs and, therefore, disposable and environmentally non-invasive energy options are generally not a viable option [81]. However, because of the miniaturized nature of in situ biosensor devices being developed, the utility of enzyme/microbial-based energy fuel cells have promising potential [55,56]. Over the past decade, several research groups have worked on the development of biosensing devices that are powered using integrated microbial fuel cells. These range from sensors for biological oxygen demand to water toxicant pollutants, e.g., formaldehyde, and provide future insight into easy to use and biodegradable energy sources [82,83,84].

The idea of real-time connectivity is a future challenge that will affect researchers working in the fields of monitoring and diagnostic devices. The idea is to establish a network of devices, both for in situ monitoring and handheld PoC devices, that provides constant real-time data instantly to centralized environmental agencies and governmental organizations. By doing so, all environmental monitoring data of, for example, microbial presence would be sent to a centralized organization that could instantly process data and return it wirelessly [74]. This would ensure a two-way system of constant monitoring data for environmental welfare groups as well as regular updates of drinking water quality to rural communities. Devices with real-time connectivity are already being explored by research groups. A group in Zimbabwe focusing on an HIV diagnostic device have integrated embedded connectivity into a prototype. Taking devices to Zimbabwe for pilot study use, they have demonstrated it is possible to significantly increase the turnaround time of external assessment testing in remote parts of Africa [85]. An option for handheld PoC devices has been presented by Smith et al. whereby a lateral flow assay device is integrated with data specific tags for picture-based data connectivity (see Figure 7) [86].

Environmental awareness is an issue that began in the early 7th century [87] and has become a central issue in modern-day society, science, and politics [88]. With the development of new technologies, during both the testing process and in commercial production, there is an ever-increasing amount of waste being produced [74]. This can include the materials used in manufacturing, reagents used during testing, devices made with unrecyclable materials, and manufacturing by-products, e.g., toxic fumes. It is clear that the future of diagnostic device development requires a strong outlook on using recyclable and sustainable materials and proper reagent disposal. Research into miniaturization of device design and reagent volume requirements will further environmental friendliness in the future. Furthermore, recent advances in paper-based sensing may provide a suitable platform for biomarker detection in the near future [47,84,86].

## 4. On-Site Biosensors, Monitoring, and Implementation of Integrated Devices in LMICs

Biosensors are devices capable of detecting biological compounds by measuring the signals given from biological recognition events to ascertain target concentrations. Biosensors are now ubiquitous within modern-day life and can be seen on the market, e.g., glucose, lactate, and Scoville sensors. Some tests can even be obtained in supermarkets, such as pregnancy tests exploiting paper-based lateral flow assays and ELISA tests with electrical signal outputs. Biosensors can be used for a multitude of applications, examples include food safety, drug detection, environmental monitoring, disease diagnostics, counterterrorism, as well as detection of pathogenic microorganisms. The general workflow of any biosensing device begins with the target analyte binding to the probe (bioreceptor) on the surface of a transducer. The transducer then translates the physical signal, e.g., heat, mass, light, etc. into an electrical signal that can be analysed and displayed to the user through the interface (see Figure 8). There are four main categories of biosensors currently described in the literature: optical, acoustic, electrochemical, and thermal [89]. This review focuses on the development of electrochemical biosensors.

### 4.1. What is an Electrochemical Biosensor?

Electrochemical biosensors are those that utilize the inherent charges of probes immobilized onto the surface of a transducer. The sensing platform can then be immersed into a solution containing charged molecules, e.g., phosphate buffer (PB) and ferro/ferricyanide ([Fe(CN)_6_]^3-/4-^). This allows for the study of electrical characteristics of various biological probe compounds on the surface as well as their interactions with other molecules. Electrochemical analysis for the development of biosensors typically uses a 3-electrode cell setup (see Figure 9) including a working electrode (WE), a reference electrode (RE), and a counter electrode (CE), or a pseudo-reference electrode accompanied alongside the working electrode in a 2-electrode system. This allows for measurement of current travelling through the system as well as the potential difference between the RE and the WE for analysis using both amperometric and potentiometric techniques. Such sensor cells are connected to a potentiostat or similar to control the input signal and record and analyse the output data.

Electrochemical biosensors, as an application of electrochemistry, have been studied extensively for their versatility, relative system simplicity, and short detection times. Electrochemical biosensors transduce the biological recognition events at the interface, i.e., the working electrode, into a processable electrical signal. This electrical signal, in turn, proportionally reflects the concentration of the analyte that has been delivered to the device. There are four main types of electrochemical biosensors seen in the literature regarding binding affinity studies, namely, amperometric, potentiometric, impedimetric, and conductimetric. Amperometric techniques involve the measurement of current relating to the oxidation and reduction potential of molecular species such as redox-capable ions in a solution. Potentiometric techniques analyse the potential variation between the RE and WE, which must be accurately controlled and stable, to ascertain the behaviour of ions in solution when effected by biological molecules on the surface of the transducer. This occurs while little to no current is present in the electrochemical cell, thus, the ion-activity data at the interface can be acquired. Impedimetric sensors allow for the analysis of complex resistance and capacitance relating to oxidation and reduction of molecular species. In impedimetric sensors, after a low-amplitude wide-range AC-frequency scan records complex impedance (real component: Z’; imaginary component: Z’’) data analysis is done by modelling impedance data with an ideal electrical circuit such as the Randles equivalent circuit, as an example (see Figure 10). This is done in order to obtain the charge transfer resistance (R_ct_) in a Faradaic process with a redox couple, or by converting the impedance into capacitance to investigate the intrinsic property of the electrochemical double layer at the interface (non-Faradaic impedance). Conductimetric techniques involve the measurement of the change of conductance for a solution after recognition of charged target analytes on the biorecognition event.

### 4.2. Advantages and Disadvantages of Electrochemical Biosensing

Electrochemical biosensors have several advantages compared with their counterparts. Advantages of electrochemical biosensing systems include low limits of detection (LOD), a wide linear response range with many target compounds, superior self-assembled monolayer (SAM) stability, and high reproducibility [90]. In this review, three different electrochemical biosensor types, that are classified based on their sensing methods, are summarized and compared in Table 1. Electrochemical sensors are widely utilized for environmental water analyses, such as handheld PH meters (potentiometric), integrated components in high-performance liquid chromatography (HPLC) (amperometric), or dedicated sensing platforms for prevalent chemical pollutant monitoring, e.g., pesticides [91], licit [92,93,94] and illicit drugs [95], endocrine disturbing compounds [96], heavy metal ions [97], etc. There is no doubt that such chemical sensors are of great importance in water quality monitoring and population size and community lifestyle estimation [98]. However, in this review, we will be discussing electrochemical biosensing applications in detecting biological markers (see Figure 11), such as proteins, oligonucleotides, and single-cell organisms [99].

Biosensors as an analytical platform provide identification and quantification of a given analyte. To enable the selectivity in a biosensor, the interface needs to be functionalized with a biorecognition element (BRE), i.e., enzymes and enzyme-labelled antibodies, antibodies, oligonucleotides, peptide aptamers, and other affinity-based non-antibody probes. Some typical sensors with immobilized BRE architecture are listed in Table 2. to provide an overview of the development of these biosensors. Interestingly, although successful biomarker detections on electrochemical interfaces have been reported for standard samples or clinical samples (i.e. serum, urine), due to the extreme extent of dilution when discharged to sewers or drainage systems, great challenges remain in detecting such biomarkers in real samples. These sensors are listed as proof-of-concept case-studies, providing that sufficiently effective preconcentration processes be implemented in the future. In matrixes as complicated as environmental water, understanding the limitation of each type of biosensor can provide a more holistic view when selecting and designing an adequate analytical interface. For example, as enzyme activities are susceptible to pH changes, enzyme-based sensors may be compromised if used for heavily contaminated water samples. Additionally, biofouling effects due to nonspecific bindings not only compromise the selectivity of affinity-based sensors but also interfere with the effective analyte diffusion in the vicinity of the working electrodes. Organic solvents or other contaminants present in the water sample may also denature the proteins and interfere in the folding of certain nucleic acid structures, i.e., aptamers.

### 4.3. How to Detect On-Site

With the rise of big data in the field of bioinformatics and the need for higher resolution monitoring analytics, the development of on-site biosensing has become a central topic. Electrochemical biosensors have the capacity to provide a truly integrated lab-on-chip (LoC) approach with a key focus on portability, capacity to achieve miniaturization, and maintaining high sensitivity (see Figure 12) [110,111,112]. 

Point-of-care (PoC) handheld or in situ monitoring devices require the integration of several components, namely, electrochemical sensing electrodes, microfluidics, signal connectivity, power sources, as well as other miniaturized elements [48,57]. The complete integration of these components would help realise the dream of a device that is capable of autonomous sample collection, preparation, testing, analysis, and data readout within the PoC, diagnostics, REASSURED, rapid-detection timeframe [73,74]. As previously mentioned, conventional laboratory techniques have the sensitivity and specificity for the detection of microbial pathogens in water. However, due to lack of rapid analysis, overall cost, and experimental complexity they are unsuitable for the purpose of on-site detection [113]. Nonetheless, the development of truly portable and autonomous diagnostic devices remains a major task. Key challenges for the implementation of these devices in an environment as complex as water will be discussed in this section. These include but are not limited to sensor integration, sample preparation [61], sample amplification/preconcentration, device connectivity [85], and antifouling [75,79].

An important justification for electrochemical sensors being a suitable candidate for environmental water monitoring in LMICs is their relative simplicity, which directly leads to cost control and convenience in device integration. One model considered suitable for on-site water detection is a self-contained portable system, called lab-in-a-briefcase, which contains the elements required for water sample collection, sample pre-treatment, biosensing elements, and data processing. The sensing component should consist of a disposable biochip and a reusable sensing device, in this case, an electrochemical cell connected to a potentiostat. Commercially available USB-powered potentiostats are an option for signal collection and data processing in such scenarios.

On the other hand, the single-use, low-cost biochip comes with more flexible variety. Currently, the available options for electrochemical disposable biochips are microfabricated electrodes on various substrates; screen-printed electrodes (SPEs); printed circuit boards (PCBs); and evaporated metals on glass slides, i.e., Au, Ag, etc. These platforms can be relatively easy to prepare in a microfabrication lab or can be custom-made through mass-production avenues [57]. The variety of fabrication methods and materials provide flexibility in design and integration. However, practical limitations for each of the platforms remain. For instance, evaporated gold electrodes on glass slides often lack mechanical strength and are sensitive to physical abrasion. An additional method to produce a preferred Ag/AgCl reference electrode may be required. However, a major drawback of SPEs and PCBs is the lack of reproducibility and purity of the metal surface during the commercial printing procedure [114].

One of the principal advantages of integrating microfluidics with electrochemical biosensors is the newfound capability of convenient sample preparation before analyte detection. Environmental water samples are relatively more complex than other samples, e.g., blood, saliva, urine, sweat, etc. due to the larger number of contaminants. Additionally, water samples vary by catchment and can contain varied contaminants based upon the surrounding catchment area and possible upstream activities, i.e., urban settlements, agricultural activities, or industrial plants. There is also the issue of analytes being found at significantly lower concentrations within these samples. With these points in mind, it has been suggested that sample preparation should include example steps such as mixing, filtration, and lysis/extraction, followed by sample preconcentration, which will be discussed further below. One of the primary reasons many PoC devices have failed to make it onto the market is due to the integration of effective sample preparation modules [115].

Some examples of micromixing techniques for reagent and sample homogenization include serpentine channel features as well as magnetic mixing using magnetized beads and sample through-flow [116]. A prototype device engineered by Lee et al. shows the fully automated gradient mixing of various solutions including measurement buffer, antibody-conjugated nanomaterials, and target antigen (see Figure 13) [117]. This microfluidic design embedded along with a gold-patterned microarray chamber for sandwich complex-based sensing achieved automated detection of multiple clinical biomarkers within 60 minutes.

Another method of mixing within microfluidic devices has been realized by the use of magnetism and magnetic beads [118]. By using a rotating magnetic disk and magnetic beads in solution it is possible to create chaotic fluid flow, thereby achieving mixing between various solutions. A study by Owen et al. implemented the use of magnetic beads in this way to mix complex analyte samples within a microfluidic device (see Figure 14), showing that the primary variables affecting efficient mixing include flow rate, rotation speed, and bead density. Using this method, it was shown that sufficient sample mixing can be achieved within relatively short distances over 5 minutes at a linear velocity of 0.19 mm/s [119].

As previously mentioned, environmental water samples are complex and thus require filtration prior to further preparation steps and detection. Multiple filtration steps may be required, ranging from the removal of macroparticles such as microplastics and humic material to removal of microparticles such as environmental DNA (eDNA) and toxicants. Recent studies show a range of ways that particle filtration can be achieved within microfluidic on-chip devices including membrane or physical filtration, magnetic bead separation, and on-chip electrophoresis [120]. Physical filtration involves the use of macro- and micromesh elements for the separation of particles based on size during constant flow. For example, simple microfluidic devices can be produced such as that shown by Qiu et al. that implemented a single nylon mesh membrane [121]. This 7-layer (see Figure 15) device contains two membranes for filtration with an inlet, waste outlet, and sample outlet for simple and fast filtration.

Magnetic beads can again be used in microfluidic sample preparation in on-chip devices for the filtration of target analytes. By functionalizing beads with analyte-complementary molecules, e.g., antigens and antibodies, it is possible to specifically target analytes of interest and remove unwanted material. An example of this includes the device shown below (Figure 16) in which a target acquisition by repetitive traversal (TART) incubator and magnetic separator are integrated. Zhou et al. have demonstrated the ability of functionalized magnetic beads to capture target analytes in a complex sample that can then be separated using a magnetic-fractionator system. Using this setup, a 90% rate of target capture and isolation was achieved, alongside a 99% removal rate of non-specific material [122]. Another less-explored method within on-chip microfluidic devices, which is steadily gaining traction, is the electrophoretic method. The advantages of microfluidic electrophoresis include low-sample requirements, cost-efficiency, rapid detection, as well as convenient integration with sensing modules [123]. Some examples of types of electrophoretic methods that have been implemented in microfluidic devices include gel electrophoresis, capillary electrophoresis, and dielectrophoresis [124,125].

The final step of PoC device sample preparation before preconcentration is sample extraction from pathogen cells within the sample, including preconcentration. The extraction of target analyte from the pathogenic microbe of interest is a four-step process: (1) cell lysis; (2) separation of target proteins, nucleic acids, etc.; (3) target purification; and (4) target preconcentration [126,127]. Steps 2 and 3 can be done using the methods of filtration and mixing discussed previously with integration of microfluidic channels and chambers. Lysis of various microorganisms, i.e., cells and viruses has seen some recent literature interest. To obtain target analytes of interest, intracellular molecules are first required to be released from within the microorganism. With the aim of extracting internal cellular components, e.g., DNA, proteins, lipids, etc., methods for miniaturization and microfluidic integration for PoC devices mostly focus on cell disruption and membrane permeabilization [128,129]. Two types of techniques exist for lysis, these are chemical lysis and physical cell disruption. Chemical lysis is currently more common and involves the disruption of the cell wall and/or membrane by denaturing proteins and permeating the lipid bilayer [128,130]. Physical cell disruption is more aggressive and involves multiple techniques including osmotic shock by exposing cells to hypotonic solutions, shearing/fracturing through microneedles or particles, electrical field lysis, ultrasonication, and thermal stress [129,131,132]. A study by Church et al. integrated the concentration and lysis of red blood cells on the same simple microfluidic chip design (see Figure 17). The device consists of an inlet and outlet joined by a 400 μm wide channel, containing at its centre a 40 μm constriction zone. Within this constriction zone, a DC-biased AC electrical field is applied to prevent leukaemia cells from entering the constriction zone. Upon the adjustment of the DC component, a simple switch from red blood cell concentration to cell lysis can be precisely controlled and performed [131].

One expected requirement for on-site diagnostic devices is the capability to autonomously preconcentrate or amplify biological sample material. This is due to the relatively low concentrations of biomarkers within water environments [133]. Additionally, in complex biological samples, sensitivity of biosensors may be decreased due to low levels of non-specific binding [134,135]. There are several different methods for the preconcentration of proteins and cells as well as the amplification of nucleic acid biomarkers used regularly within laboratory settings. For transference to on-chip diagnostics, there exist a few protein preconcentration options including solid-phase extraction (SPE) [95,136], affinity-based magnetic bead enrichment [137], and isoelectric focusing (IEF) [138]. For nucleic acid amplification, several techniques are being explored for miniaturization using integrated platforms and microfluidics. Examples include polymerase chain reaction (PCR), loop-mediated isothermal amplification (LAMP), and nucleic acid sequence-based amplification (NASBA) among others [139].

In protein biomarker detection, due to low concentrations of analyte in samples and the reality of limited electrode surface area, a relatively small sample volume and thus analyte amount is present. One of the solutions to realise a relatively rapid on-site measurement may require the combination of a preconcentration process and microfluidic assay. Three methods are considered applicable for further protein enrichment in the filtered sample: (1) SPE [95] using gel loaded with affinity-based chemical capturing molecules or biomimetic polymer [140]; (2) affinity-based magnetic beads-based protein enrichment [136]; and (3) IEF, using electrophoretic principles for separation and preconcentration by their isoelectric point (PI). Currently, such methods likely present a risk of losing analyte molecules in a flow-through solution. Another imposed challenge is the lack of portable power supply to support centrifuge, vacuum, and microflow for the filtration process should these be needed. As previously mentioned, lack of power sources has already been identified as one of the infrastructural challenges not only for conventional techniques, but also PoC device implementation in developing countries [137].

Solid-phase extraction (SPE) is often compared with chromatography. However, within PoC devices it is a purification method for samples of given analytes, rather than a separation method for all compounds in a complex sample. SPE implements a flow-through container/column (i.e., cartridge, disc, etc.) that is embedded with target-specific modified molecules. A sample of the environmental water is incubated with the conditioned molecule for a period of time then rinsed with washing buffer to eliminate the non-specific compounds. Subsequently, the intended analytes are extracted in elution buffer for further analysis. Flow-through or extraction is usually performed with the assistance of gravity or positive pressure, such as vacuum or microcentrifugation. Modern analytical chemistry has been developed to accommodate highly selective small volume assays, however, this feature may not be sufficient in preconcentrating extremely diluted samples such as environmental water. When isolating specific proteins or biomolecules from environmental water, to achieve the concentration that reaches the linear range of detection of a biosensor, litres of grab samples may need to be processed. Current SPE devices, such as large-reservoir cartridges (LRC), can only handle sample volumes of up to 150 mL [141]. One way to overcome this hurdle is to use a combined series of extraction/purification procedures, which does, however, conflict with the aim of realizing a simple, rapid on-site and low-cost sample-processing device for biomarker monitoring. As for magnetic separation, Figure 18 shows a time-efficient and relatively simplistic way to concentrate specific protein analytes from a complex matrix [142]. This technique can be applied to the purification of proteins, cells, or other biomolecules [143].

An alternative method of protein preconcentration is isoelectric focusing (IEF). This is a steady-state electrophoretic method that simultaneously separates and concentrates proteins at their isoelectric point (PI) [138,144], the pH at which a protein is neutrally charged. This method also works with peptides. IEF is achieved by applying an electric field to a system containing a sample and a stable pH gradient. When an electric field is applied to the system, proteins within the sample will migrate towards either the cathode or anode depending on the proteins’ charge. Once a protein reaches its PI, it will settle during a period of stabilization, hence separating the proteins in the sample and concentrating them in bands at their respective PIs (see Figure 19).

Methods of pH gradient formation often include the use of carrier ampholytes, either immobilized in the form of a gel or free in solution [138]. Broad range carrier ampholytes in solution form a pH gradient when exposed to an electric field [145]. Ampholytes are an example of a molecule with at least one acidic and one basic acid dissociation constant (PKa), another example being a zwitterion [146]. IEF is capable of being translated into microfluidic platforms. The advantages of the miniaturization of IEF into chip-based devices include, but are not limited to, reduced analysis time and cost and low sample volumes. A recent example of IEF in a microfluidic device is reported by Yu et al. [147]. A paper-based analytical device (μPAD) was successfully used to separate and preconcentrate a sample including myoglobin and cytochrome C (see Figure 20). According to Yu et al, a separation resolution larger than 1.5 indicates efficiency; a separation resolution of 3.5 was achieved with the μPAD. In comparison, an equivalent gel-based IEF system gave a resolution of 3.7, which indicates an effectively miniaturized device.

Similar to protein and cell monitoring, due to the low concentrations of nucleic acids present in environmental samples, it is expected that nucleic acid-based biosensing systems will not be sensitive enough. The most common form of nucleic acid amplification is PCR [148]. PCR is a technique that was invented by Kary Mullis in 1985, allowing scientists to amplify small sample sizes to create millions of copies of an isolated DNA sequence [149]. There is now a multitude of different takes on the original technique of thermocycling DNA samples between three temperatures with added enzymes and nucleotide bases (see Figure 21) [150].

Over the last decade, a lot of research has focused on the development of electrochemical μPCR on-chip devices [151,152]. In order to miniaturize the PCR process, devices have utilized microfluidics coupled with miniature heating elements made of metals using various substrates, e.g., PCB, silicon, glass, etc., and sensing electrodes [153]. One such device was developed by Moschou et al. (see Figure 22), in which PCB was used to integrate DNA amplification with subsequent detection on a Si-based sensing array. Serpentine microfluidics were placed over resistance-based microheater elements of varying set temperatures where the sample undergoes 30 cycles of PCR. The amplified product then enters the sensing chamber to be detected through a change in capacitance due to hybridization [154].

Another report from Hwang et al. shows the use of propylene and 400 μm double-sided tape in place of complex microfluidic layers to produce a relatively simplistic channel for the same process as described above. These channels were placed over that of a PCB-based μPCR platform containing the heating circuit and thermal sensor. Electrophoresis results showed similar amplification performance to the conventional thermal-cycler PCR method [155]. Loop-mediated isothermal amplification (LAMP) is another promising technique to quickly preconcentrate nucleic acids developed by Notomi et al. in 2000 [156]. The advantage of LAMP over traditional PCR methods is the completion at a single set temperature (approximately 65 °C) within a single tube [157,158]. It involves a three-step process starting with sample material priming, amplification, and then elongation and further exponential amplification. A recent study by Ma et al. developed the first easy-to-use, automated device for sample processing, amplification by LAMP, and detection of bacterial species. The device is a smartphone-controlled, passive microfluidic, colourimetric-based device and integrates sample filtering, cell lysis, LAMP, and a colourimetric sensor (see Figure 23). Sample collection to data output was observed in 40 minutes with detection limits of 3.2 × 10^−3^ HAU or 30 CFU per reaction of *Staphylococcus aureus* [159].

## 5. Conclusions and Future Perspectives

In this review, we have given an overview of the current challenges within the field of environmental water quality monitoring and the possible implementation of integrated electrochemical biosensing devices. We have discussed the primary issues that the developing world is facing regarding the prevalence of waterborne pathogen-related disease. Also discussed are the conventional techniques that are currently used in the developing world and their associated strengths and weaknesses. Challenges for the implementation of on-site diagnostics have been outlined, including ease of use; the cost of manufacturing and implementation; capability of in situ water quality analysis; and collection of real-time qualitative data, amongst others. Electrochemical biosensors are currently a promising avenue of environmental water diagnostic devices. This is due to their inherent ability for miniaturization and easy integration, low LODs, wide linear response range, superior electrode surface stability, high specificity, and low-power equipment requirements. Advantages and disadvantages of current electrochemical biosensing techniques are discussed. Examples of various integration modules for on-site electrochemical devices have been demonstrated within the literature. Examples of mixing such as magnetic beads and serpentine microfluidic channels were shown. Possible techniques for microfiltration units include physical mesh and membrane filters, magnetic bead fishing techniques, as well as electrophoretic methods. We also reviewed examples of analyte extraction technology that have been developed in the literature, implementing chemical lysis and physical disruption methods. Examples of chemical lysis include use of microfluidic channels and chambers with detergents and chaotropic reagents and show promising capability. Physical disruption techniques are arguably more proof-of-concept and use osmotic shock; microneedles and particles for shearing and cell fracturing; electrical fields for cell membrane/wall disruption; and thermal on-chip lysis.

While examples of various easily integrated modules independently exist within the current literature, very few examples exist of these modules being integrated into a lab-on-chip or lab-in-a-briefcase type device. The next steps for researchers is to work on the integration of some, and eventually all, of these components to realize their full potential in a self-contained device. This remains a challenge, however, due to the nature of these devices being developed separately thus far. It is suggested that those attempting to integrate individually-developed components may experience compatibility issues in doing so. Consequently, the future development of individual modules requires the mindset for components to be capable of attaching to platforms such as PCBs, SPEs, and thermally deposited glass devices. Integration with potentiostats, various electronic equipment, and low-power sources must also be considered. It is worth mentioning in the spirit of environmental friendliness that future development of these components should be done using as many recycled, recyclable, and sustainable materials as possible. Environmental water samples are complex, more complex than most biological samples, containing many hundreds of potential biomarkers. When looking at physiologically relevant markers for waterborne pathogens, this can be reduced to 10–20 biomarkers. Therefore, it is also important that future development of electrochemical devices to detect such pathogens include multiplexing.

With all of the above in mind, the future goals of electrochemical biosensing devices remain the same. The development of fully integrated autonomous biosensor systems is the key goal of those developing devices for environmental water quality monitoring. This would enable all countries to produce constant long-term sampling data for the bioinformatics field. This would permit higher-resolution trends in overall water quality and waterborne pathogen outbreaks with direct data feedback connectivity. A holistic approach is advised, involving research individuals from not only the STEM disciplines but also the social sciences and members of economic and political groups. This is directly important to ensure technology development and implementation in the developing world with members of the public and local communities in mind.

## Figures and Tables

**Figure 1 biosensors-10-00036-f001:**
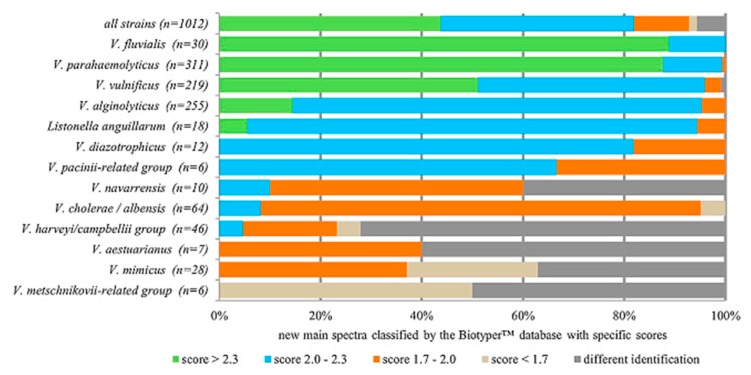
Classification of new main spectra with the Biotyper™ database. The highest matching scores of the new main spectra were divided into ranges reflecting highly probable species identifications (>2.3), probable species identifications (2.0–2.3), secure genus identifications (1.7–2.0), and unreliable identifications (<1.7). Strains with Biotyper™ species classifications are different from the species assignment for VibrioBase main spectra. (Re-printed from Erler et al. [28]. Copyright (2015), with permission from Elsevier).

**Figure 2 biosensors-10-00036-f002:**
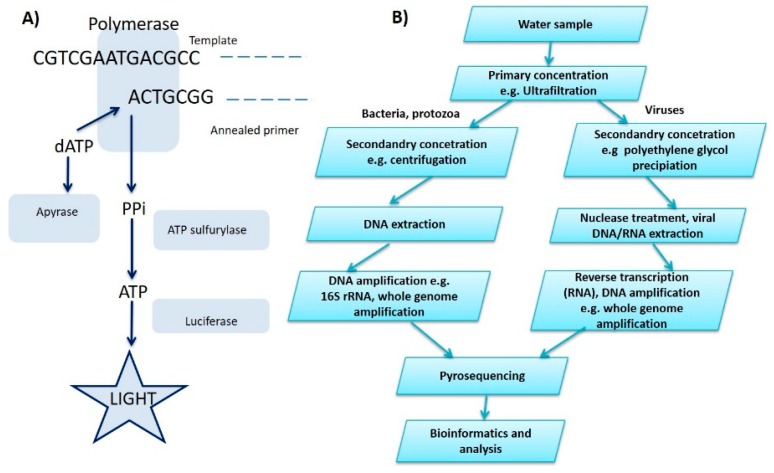
(**a**) Schematic illustration of the pyrosequencing reaction with four enzymes (DNA polymerase, ATP sulfurylase, luciferase, and apyrase). Nucleotides are added one at a time to form the complementary strand of the single-stranded template, to which a sequencing primer has been annealed. Each nucleotide incorporation event is accompanied by the release of pyrophosphate (PPi). ATP sulfurylase converts the PPi into ATP. The ATP is then converted to visible light by luciferase and the produced light signal is detected. Unincorporated nucleotides and ATP are degraded by apyrase between each cycle. (**b**) Major steps in the metagenomic detection of microbial pathogens in water by pyrosequencing. (Modified from Aw and Rose [38] Copyright (2011), with permission from Elsevier).

**Figure 3 biosensors-10-00036-f003:**
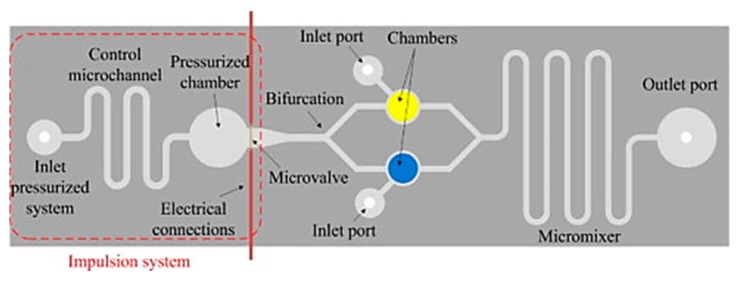
Schematic for a portable microfluidic platform composed of printed circuit board (PCB) technology with SU-8 photoresist. Schematic shows the impulsion system containing inlet, automated microchannels, and pressurized chambers combined with electronics followed by micromixer serpentine design and sample outlet. (Re-printed from Aracil et al. [61]. Copyright (2015), with permission from Elsevier).

**Figure 4 biosensors-10-00036-f004:**
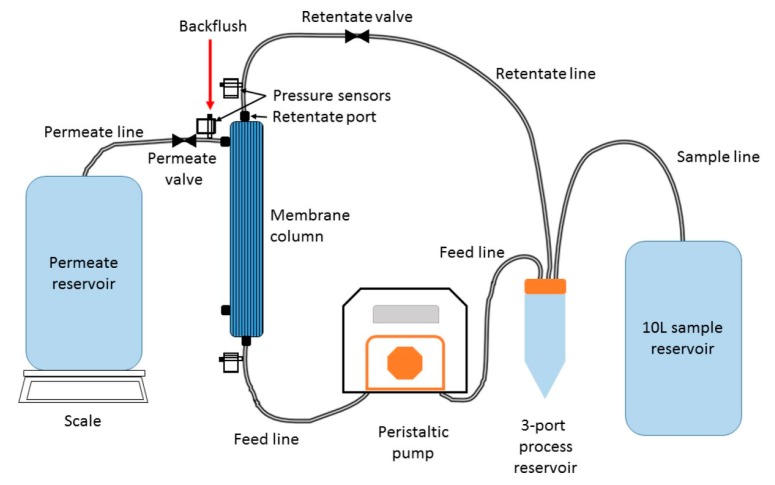
Schematic depiction of the experimental setup used for step one of the two-step preconcentration process for virus recovery from fresh and wastewater samples. (Re-printed from Farkas et al. [67]. Copyright (2018), with permission from MDPI AG).

**Figure 5 biosensors-10-00036-f005:**
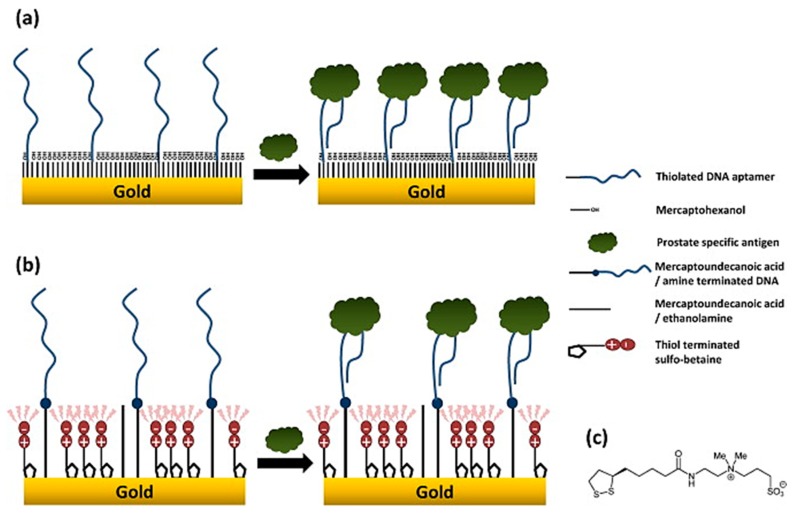
Self-assembled monolayers (SAMs) with DNA aptamers on a gold surface before and after incubation with PSA. (**a**) Thiolated aptamer with MCH; (**b**) amine-terminated aptamer with sulfo-betaine; (**c**) structure of the thiol-modified sulfo-betaine. (Re-printed from Jolly et al. [79]. Copyright (2015), with permission from Elsevier).

**Figure 6 biosensors-10-00036-f006:**
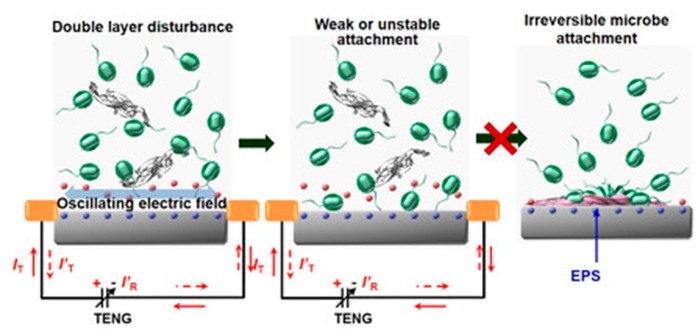
Schematic of a proposed anti-biofouling mechanism. Where an oscillating electric field exists, the surface double layer was disturbed leading to unstable organic and microbe attachment. (Re-printed from Long et al. [80]. Copyright (2019), with permission from Elsevier).

**Figure 7 biosensors-10-00036-f007:**
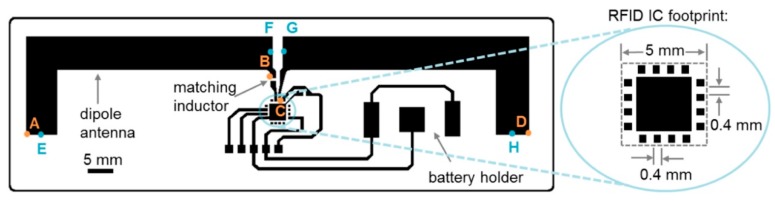
Tag design (110 mm × 35 mm) based on a development kit for printing onto different substrates. A to B and C to D show the longest path over which resistance measurements are made for each antenna arm, while E to F and G to H show the centre paths over which resistance is measured. (Re-printed from Smith et al. [86]. Copyright (2018), with permission from IOP Publishing Ltd.).

**Figure 8 biosensors-10-00036-f008:**
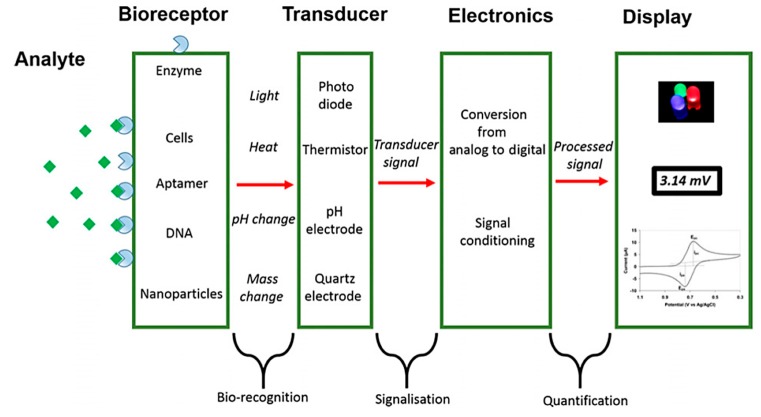
Standard components of a biosensing device. (Re-printed from Bhalla et al. [89]. Copyright (2016)).

**Figure 9 biosensors-10-00036-f009:**
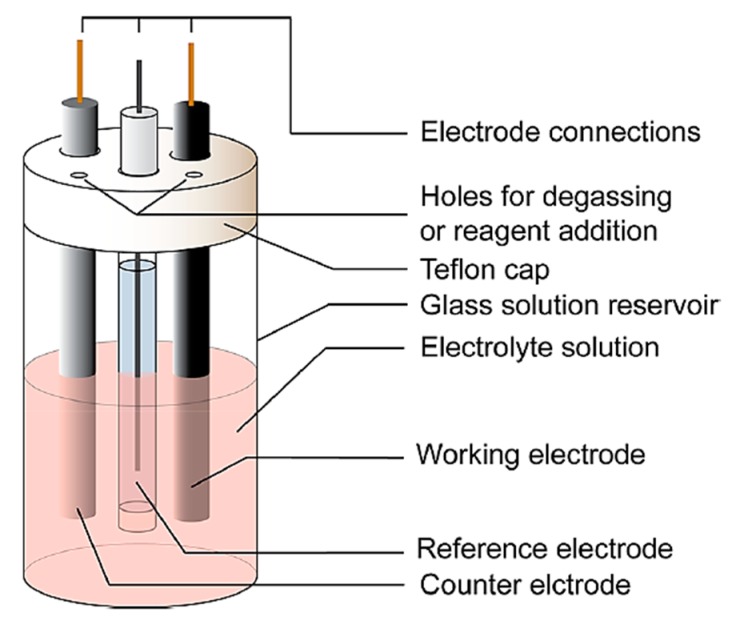
Typical 3-electrode cell setup with a working electrode usually made of gold, a reference electrode, which may be Ag/AgCl, Hg/HgSO4, or SCE, and a thin-wire Pt counter electrode.

**Figure 10 biosensors-10-00036-f010:**
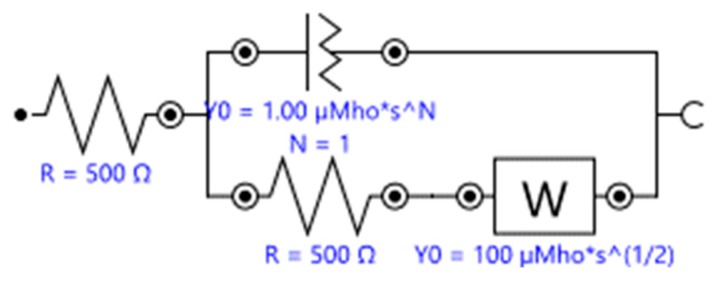
Randles equivalent circuit.

**Figure 11 biosensors-10-00036-f011:**
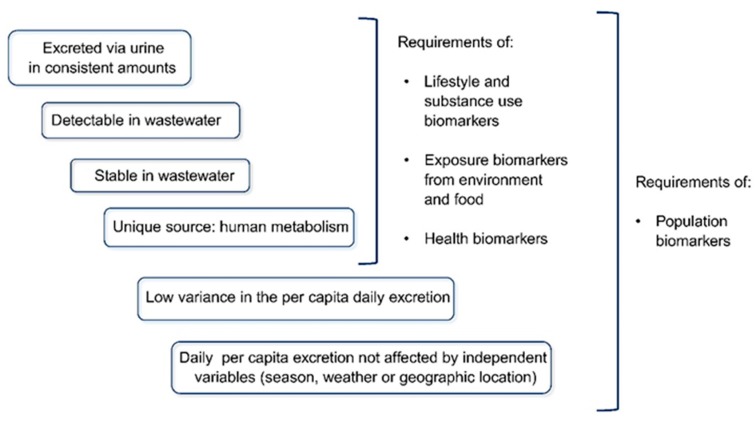
Guidelines for selecting a biomarker in wastewater diagnostics. (Re-printed from Gracia-Lor et al. [99]. Copyright (2017), with permission from Elsevier).

**Figure 12 biosensors-10-00036-f012:**
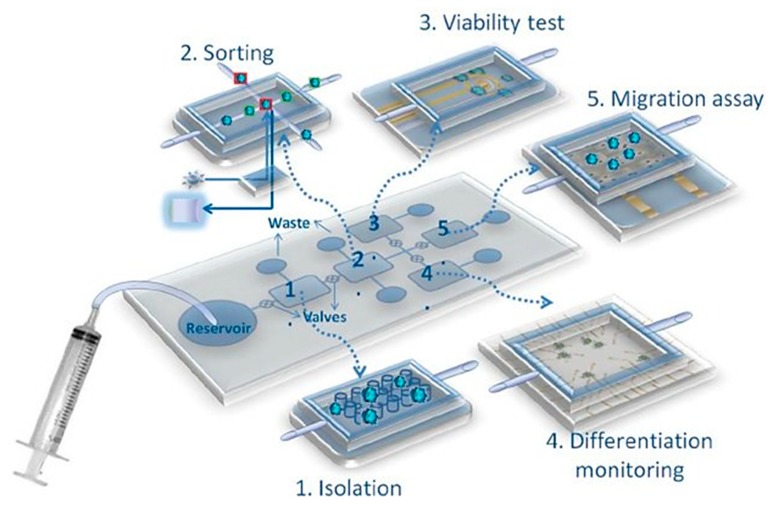
Example of a modular lab on a chip for stem cell studies. Several microfluidic components and sensing modules are integrated together for cell isolation, detection and counting, viability or migration assays, and differentiation studies. (Re-printed from Primiceri et al. [111]. Copyright (2013), with permission from The Royal Society of Chemistry).

**Figure 13 biosensors-10-00036-f013:**
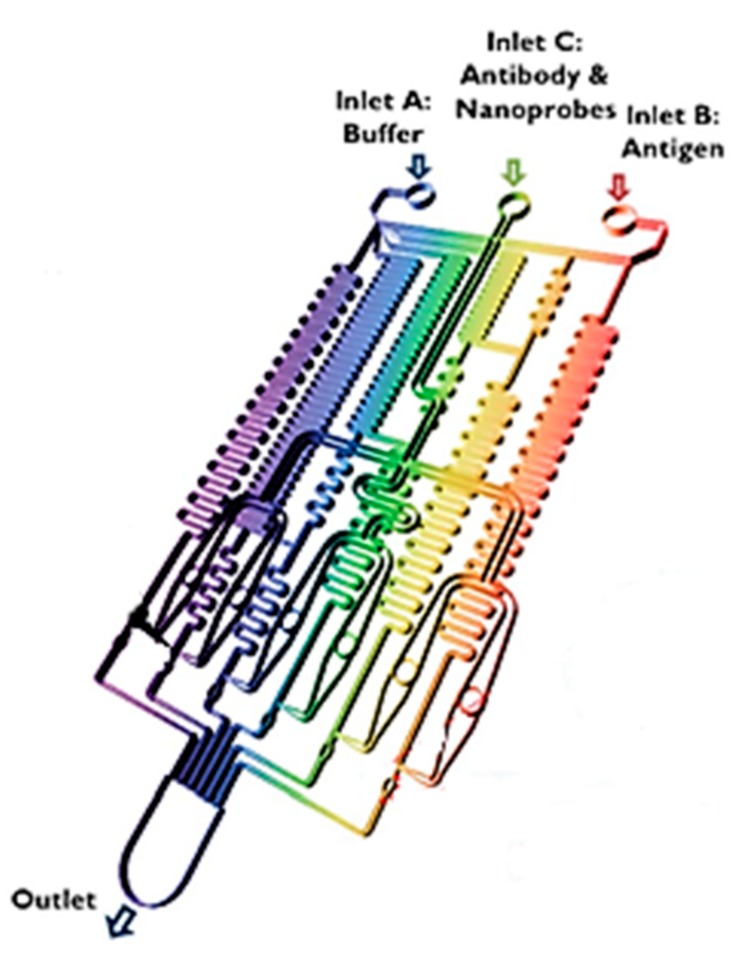
Layout of the gold array-embedded gradient chip. (Re-printed from Lee et al. [117]. Copyright (2020), with permission from The Royal Society of Chemistry).

**Figure 14 biosensors-10-00036-f014:**
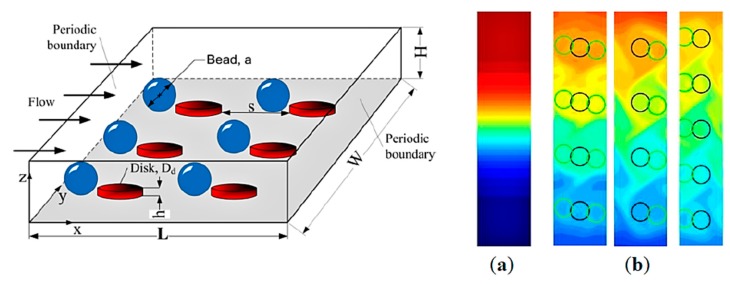
(**left**) Computational setup of the magnetic mixer. (**right**) Comparison of mixing efficiency between pure diffusion (**a**) and magnetic mixer without channel flow after 100 bead revolutions (**b**). The concentration is averaged over channel height. The back circles denote the position of static discs, whereas the green circles indicate the instant position of rotating beads. In (**b**), left, 2 beads rotation per disk with 2a spacing; middle, 1 bead per disk with 2a spacing; right, 1 bead per disk with 1.5a spacing. (Re-printed from Owen et al. [119]. Copyright (2013), with permission from MDPI AG).

**Figure 15 biosensors-10-00036-f015:**
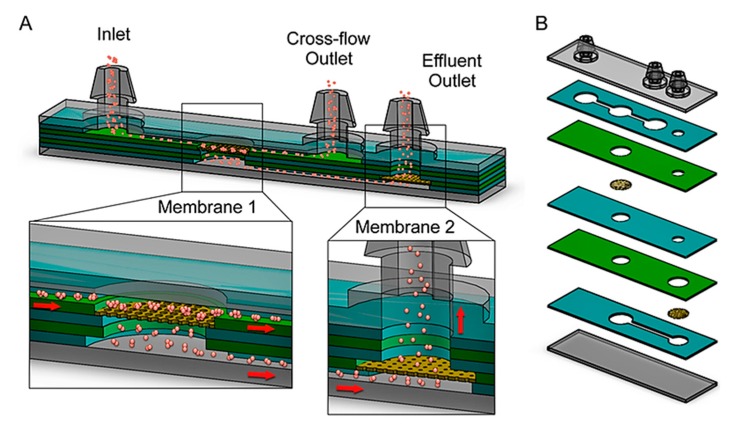
Microfluidic filter device for tissue specimens. (**a**) Schematic of the microfluidic filter device containing two microporous membranes. The first membrane is located in the centre of the device and is intended to restrict large tissue fragments and aggregates from passing through to the effluent outlet (direct filtration). If desired, some of the sample can be passed over the surface of the first membrane for collection from the crossflow outlet (tangential filtration). The second membrane is immediately upstream of the effluent outlet and is intended to restrict smaller aggregates from reaching the effluent outlet. (**b**) Exploded view showing seven PET layers, including three channel layers, two via layers, and two layers to seal the top and bottom of the device. (Re-printed from Qiu et al. [121]. Copyright (2018), with permission from The Royal Society of Chemistry).

**Figure 16 biosensors-10-00036-f016:**
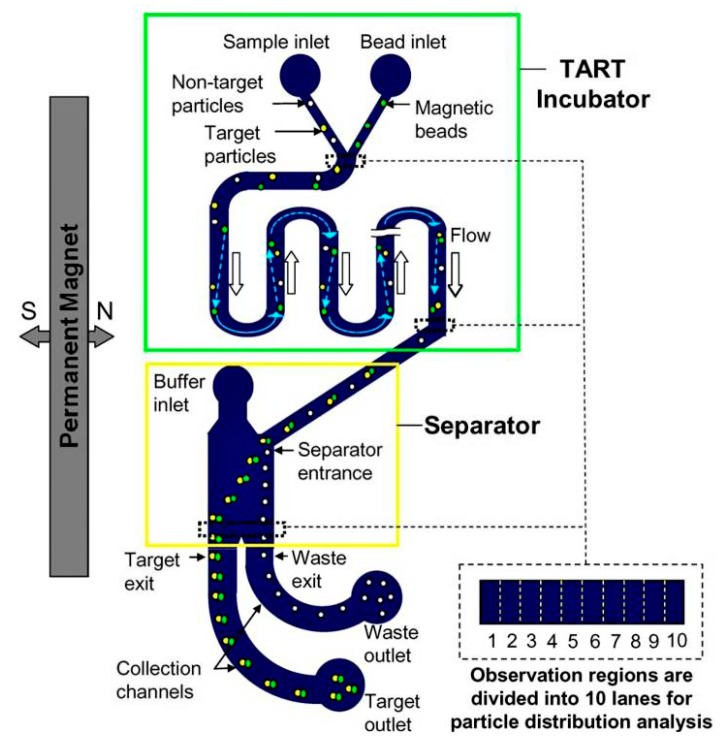
Schematic design of the microfluidic device for continuous-flow magnetically-controlled capture and separation of microparticles. The device consists of an incubator that employs the target acquisition by repetitive traversal (TART) of magnetic beads and a separator that uses magnetic fractionation. The incubation and separator are serially connected and placed next to a bar-shaped permanent magnet. Particle distributions are analysed in the observation regions at the Y-junction, the end of the incubator, and the end of the separator. These regions are each divided into ten lanes to facilitate the analysis. (Re-printed from Zhou et al. [122]—permission pending).

**Figure 17 biosensors-10-00036-f017:**
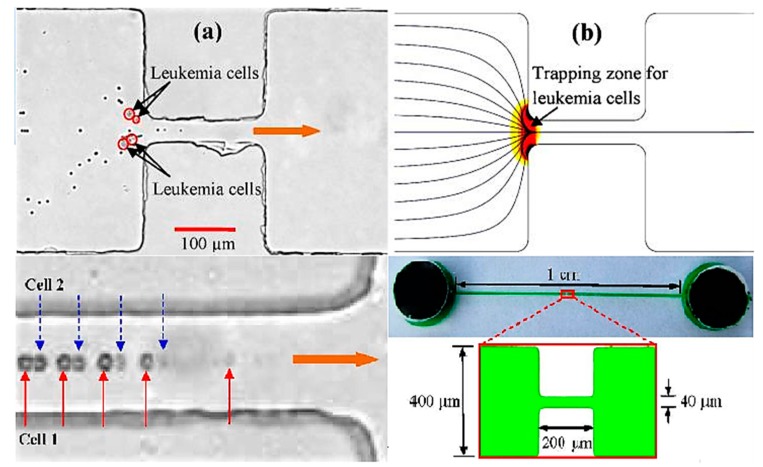
(**top**) Integration of electrical concentration of leukaemia cells and electrical lysing of red blood cells in the microchannel constriction: (**a**) experimentally recorded snapshot image with the trapped leukaemia cells highlighted with circles for clarity and (**b**) numerically predicted trapping zone for leukaemia cells and trajectories for red blood cells. The block arrow indicates the flow direction. (**bottom left**) Superimposed image of two red blood cells (highlighted as cell 1 and cell 2) illustrating the typical process of electrical lysis in the microchannel constriction. The block arrow indicates the flow direction in the channel. (**bottom right**) Picture of the microfluidic chip and dimensions of the constriction microchannel. (Re-printed from Church et al. [131]. Copyright (2010), with permission from The American Institute of Physics).

**Figure 18 biosensors-10-00036-f018:**
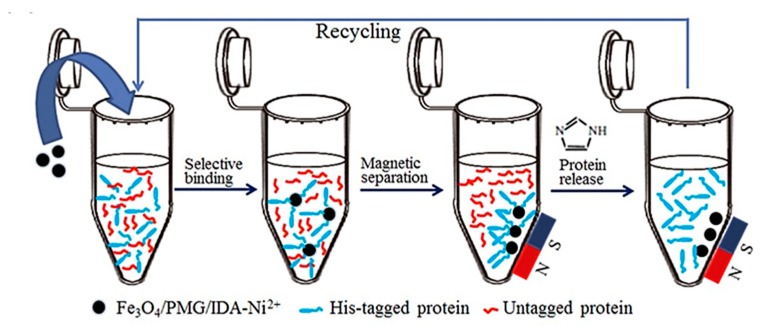
A simplified example of magnetic protein separation using functionalized-magnetic beads in a single tube. (Re-printed from Zhou et al. [142]. Copyright (2018), with permission from Elsevier).

**Figure 19 biosensors-10-00036-f019:**
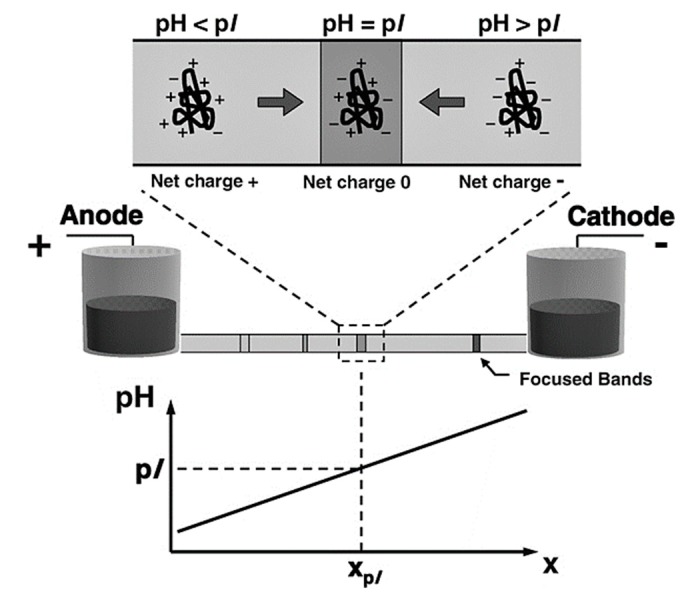
IEF concept schematic. Amphoteric molecules are driven along the pH gradient towards their PI. (Reprinted from Sommer and Hatch, 2009 [138], with permission from Wiley-VCH Verlag GmbH & Co. KGaA, Weinheim).

**Figure 20 biosensors-10-00036-f020:**
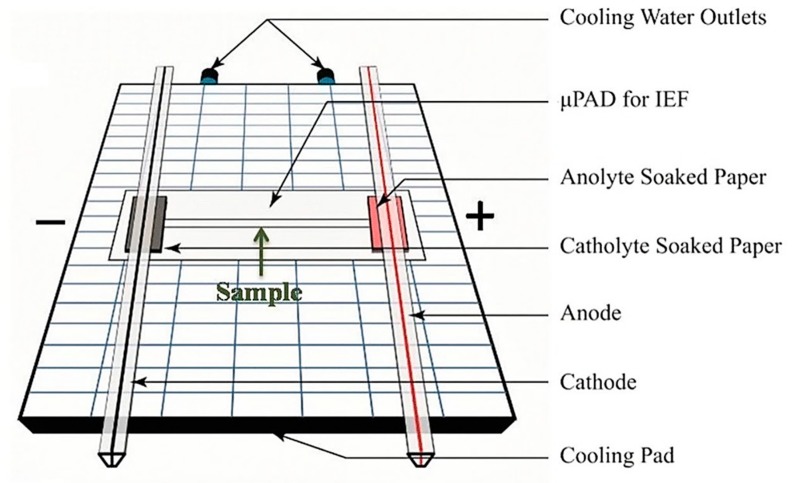
Schematic design of the μPAD used by Yu et al. The device was placed on a cooling pad and the connections were fixed to the anolyte and catholyte soaked paper. (Re-printed from Yu et al. [147], with permission from Springer Nature).

**Figure 21 biosensors-10-00036-f021:**
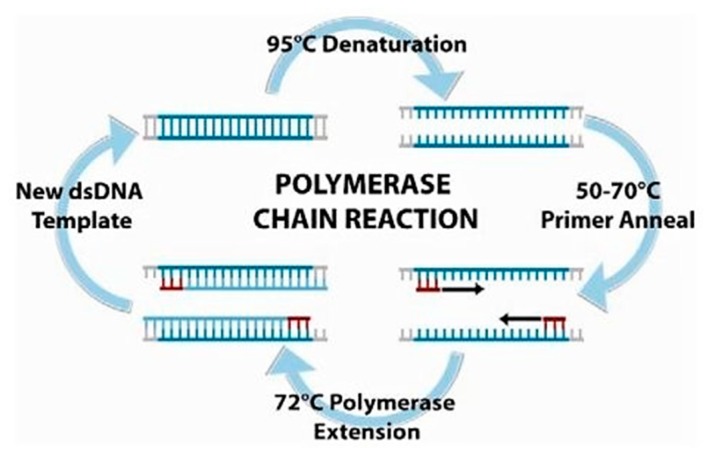
Polymerase chain reaction (PCR) standard protocol involves raising the temperature of the reaction to 95 °C to separate the DNA strands, lowering it to the annealing temperature for the oligonucleotide primers to hybridize, and then raising it to the optimal DNA polymerase temperature of 72 °C for primer extension. This process is repeated cyclically, creating exponential copies of the target sequence. (Re-printed from Lui et al. [150]. Copyright (2009), with permission from MDPI AG).

**Figure 22 biosensors-10-00036-f022:**
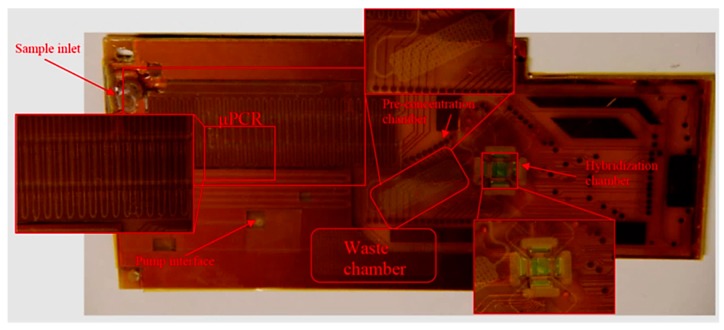
Photograph of the fabricated PCR-on-chip device. (Re-printed from Moschou et al. [154]. Copyright (2013), with permission from Society of Photo-Optical Instrumentation Engineers (SPIE)).

**Figure 23 biosensors-10-00036-f023:**
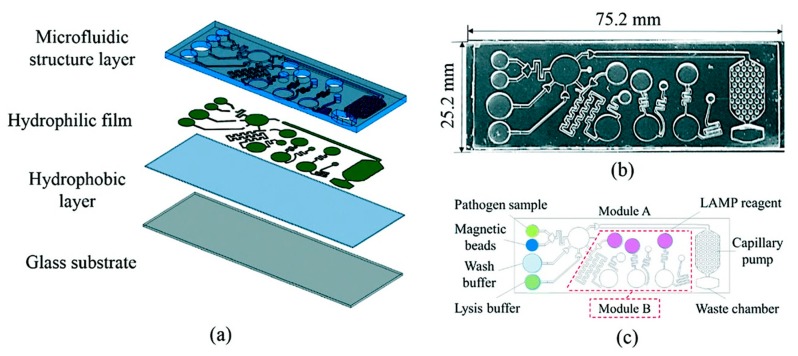
(**a**) Exploded view of the self-driven microfluidic chip consisting of a polydimethylsiloxane (PDMS) microfluidic structure layer, a hydrophilic film, a PDMS hydrophobic layer, and a glass substrate. (**b**) A photograph of the chip. (**c**) Schematic illustration of the chip design. The area enclosed by the red dotted line represents the LAMP reaction module (module B), with the other area being used for sample treatment (module A). (Re-printed from Ma et al. [159]. Copyright (2019), with permission from The Royal Society of Chemistry).

**Table 1 biosensors-10-00036-t001:** Electrochemical biosensor classifications and their advantages and disadvantages in environmental water monitoring.

	Amperometric (Voltammetric)	Potentiometric	Impedimetric
**Input**	Controlled constant voltage or a controlled series of voltages (e.g., linear potential sweep, cyclic voltammetry, differential pulse voltammetry, etc.).	A defined voltage at the reference electrode. Little to no current presence.	A DC bias potential and a wide range AC frequency scans (Faradaic impedance) or wide range AC frequency scan (non-Faradaic capacitance).
**Output**	Change in current against time or change in current against the applied voltages.	The potential difference between the WE and the RE (a constant potential).	Complex impedance or complex capacitance.
**How to Improve**	Utilize electron mediator or conductive nano-architectures (e.g., nanotubes) or employ interdigitated electrodes to amplify the signal and to improve the signal-to-noise ratio.	Incorporates CNT into the sensor to improve heterogeneous charge transfer and increase surface area [100].	Immobilize nanoparticles as a secondary architecture to increase the surface area and improve charge transfer [101]. For capacitive sensors, implement single frequency monitoring to realize a faster and simpler system.
**Principle of Detection**	Measured Faradaic current is proportional to the concentration of the analyte.	The potential difference between the WE and RE is proportionally present in the ion activity in the sample. Widely applied with ion-selective electrodes (ISE).	Measured complex impedance contains the information of charge transfer resistance (if redox couple is present) and the capacitive property of the electrochemical double layer in the vicinity of the WE.
**Advantages**	System simplicity (easy to integrate and potentially low cost); fast detection [102]; Low detection limit and high sensitivity [103]; potential in intercalator based DNA detection using PNA probes [104].	Suitable for mass-production.	Potentially label-free and easy to prepare the electrode.
**Limitations**	pH changes upon the introduction of a biological specimen may impede the enzyme activity.	High requirement to the reference electrode (stable and accurate) [102]; same drawbacks of using enzyme as a label or a BRE; inherently susceptible to the environment’s pH value during the measurements as the detecting method relies on the ion activity; limited applications due to difficulties in being employed in affinity-based detections.	Taking relatively more time to conduct one measurement due to the time-consuming frequency scan process; difficulties in miniaturization due to the more complicated input and output control; for its label-free nature, an adequate antifouling system and certain signal amplification modifications may be required.

**Table 2 biosensors-10-00036-t002:** Examples of proof-of-concept electrochemical biosensors.

Analyte	Analyte Type	Biosensing Method	Biorecognition Element	Labelling	Detection Limit (sample matrix)	Ref.
***Salmonella typhi***	DNA sequence	Voltammetry (differential pulse voltammetry)	Complementary DNA Probe	Methylene blue (MB) redox intercalator	10 fM within 60 s hybridisation time, 100 fM in serum samples	[105]
***Escherichia coli***	Whole cell sandwich assay	Cyclic voltammetry	*E. Coli* surface antibody	Horseradish peroxidase antibody (electrochemical sandwich ELISA)	50 cfu mL^−1^ in water samples	[106]
**methicillin-resistant staphylococcus aureus (MRSA)**	DNA sequence	Electrochemical impedance spectroscopy	DNA sequence	Label-free (in presence of ferrocene redox mediator)	100 fM in buffer	[107]
***Escherichia coli*** **(O157:H7 serotype)**	DNA sequence	Electrochemical impedance spectroscopy	DNA sequence	Label-free	1 fM in buffer	[108]
**Yeast cells**	Whole cell microbe	Chronopotentiometry	Molecularly imprinted polymer (MIP)layer	Label-free	Cultured yeast cells: 50 CFU/ml (diluted in buffer)	[109]

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
