# Peer review of "Integrated Electrochemical Biosensors for Detection of Waterborne Pathogens in Low-Resource Settings"

_biosensors, 2020, doi:10.3390/bios10040036_

Round 1
Reviewer 1 Report
The authors have nicely and thoroughly revised the challenges and opportunities involved in the quality monitoring of environmental water. After providing a detailed list of the requirements to guarantee proper water monitoring in low and middle income countries, they have described the advantages and limitations of the use of electrochemical biosensors to fulfil those needs. Special focus has been on the implementation of these devices in on-site monitoring, considering all the reported possibilities to address their integration.
The paper is nicely written, and my only recommendation would be to revise section 4, especially the last part (4.3) devoted to on-site detection to minimise repetition of the description of some of the approaches suggested to achieve a fully integrated device.
Although I understand that the main purpose of the authors is to provide the readers with a wide range of possibilities for developing integrated electrochemical biosensors suitable for water quality monitoring, I found a bit distracting the fact that most of the examples included refer to other types of applications. This is ok when no examples for the detection of water pathogens exist, but as an example that I recommend to address, Table 2 which includes examples of proof-of-concept electrochemical biosensors could be rewritten only including examples that are relevant for water quality applications.
Section 2.5 about pyrosequencing describes this sequencing method, but unlike all the other conventional techniques included in section 2, does not include proper discussion of its advantages (it is only said that looks promising to speed up and reduce costs) and disadvantages.
I also have a few minor comments.
Page 5, line 220: Samples for SEM analysis are often coated with a thin film of gold, so “heavy metal” should be changed by just “metal”.
Page 7, line 285: Staphylococcus aureu is missing the “s” of aureus.
Page 15: Conductometric sensors should also be named as a fourth type of electrochemical sensors.
Page 15: EIS data is not always fitted using a Randles equivalent circuit.
Page 15, line 591: Instead of “with ferrocene based redox couple” should be “with a redox couple” to generalise.
Page 17, line 629: Instead of “comprised” should read “compromised”
Page 20, Figure 13: Similar designs including micromixing have been developed for electrochemical sensing, so I would include one of those examples rather than a SERS-based platform.
Page 27, line 910: What does HAU refer to? Please indicate which type of bacterial species were quantified using the described approach.
The review covers an interesting topic, providing useful information for researchers working on this topic, so I recommend publication of the manuscript after minor revision.
Reviewer 2 Report
Although the title is “integrated electrochemical biosensors for …”, actually the authors include many different kinds of biosensors and analytical methods in the manuscript. Meanwhile, not only biosensors, but also many normal analytical methods were described. In generally, the reviewer think that the present content of the manuscript is too broad to be handled in single manuscript. The authors should focus it on a more special field.Author Response
Respectfully, we disagree with the Reviewer. We feel that there are already several review papers tackling specific aspects of environmental sensors. We wanted to provide a more holistic overview of the current trends and future directions of this area of interdisciplinary and integrative research. We strongly believe that such a broad-value point of view is useful to the community working in this area or considering working in this area.
Reviewer 3 Report
I found some typing errors. The L. monocytogenes is reported well, but twice you wrote monmocytogenes
at line 285 Staphylococcus aureu.
814 that embedded is / is embedded
I'd like to suggest to quote also Chiriacò et al. Impedance Sensing Platform for Detection of the Food Pathogen Listeria monocytogenes. Electronics 2018, 7, 347; doi:10.3390/electronics7120347
overall, the overview of future easy-to-use portable devices is complete. There is a problem with immuno-based systems, since we need a wide range of antbodies or MIPs to capture all the analytes that may be present in a sample, and wastewater may contain a great set of them. however, when pathogens are concerned, this number can be reduced to a set of 10-15.
Round 2
Reviewer 2 Report
The reviewer still think that the content of this manuscript is too broad to be suitable for a normal reveiw paper.